# MgAl-Layered-Double-Hydroxide/Sepiolite Composite Membrane for High-Performance Water Treatment Based on Layer-by-Layer Hierarchical Architectures

**DOI:** 10.3390/polym11030525

**Published:** 2019-03-20

**Authors:** Zongxue Yu, Xiuhui Li, Yixin Peng, Xia Min, Di Yin, Liangyan Shao

**Affiliations:** 1College of Chemistry and Chemical Engineering, Southwest Petroleum University, Chengdu 610500, China; lixiuhui727@163.com (X.L.); maoxiangxin888@163.com (Y.P.); 15002895606@163.com (X.M.); m15708484756@163.com (D.Y.); 17683164120@163.com (L.S.); 2Oil & Gas Field Applied Chemistry Key Laboratory of Sichuan Province, Southwest Petroleum University, Chengdu 610500, China; 3State Key Laboratory of Oil & Gas Reservoir Geology and Exploitation, Southwest Petroleum University, Chengdu 610500, China

**Keywords:** MgAl-LDH/Sep, 3D hierarchical structure, cationic dye, anionic dye, hydrophilicity

## Abstract

One of the major challenges in the removal of organic pollutants is to design a material with high efficiency and high flux that can remove both cationic and anionic dyes, oil-in-water (O/W) emulsion and heavy metal ions. Herein, we constructed novel chemically stabilized MgAl-layered-double-hydroxide/sepiolite (MgAl-LDH/Sep) composite membranes via 3D hierarchical architecture construction methods. These membranes were analyzed by scanning electron microscopy (SEM), atomic force microscopy (AFM) and X-ray diffraction (XRD), etc. Benefiting from the presence of hydrophilic functional groups on the surface of the film, the membranes show an enhanced water flux (~1200 L·m^−2^ h^−1^), while keeping a high dyes rejection (above 99.8% for anionic and cationic dyes). Moreover, the CA membrane coupled with MgAl-LDH/Sep exhibits a multifunctional characteristic for the efficient removal of mesitylene (99.2%), petroleum ether (99.03%), decane (99.07%), kerosene (99.4%) and heavy metal ion in water due to the layer-by-layer sieving. This hierarchical architecture is proved to have excellent environmental and chemical stability. Therefore, the membrane has potential in the treatment of sewage wastewater.

## 1. Introduction

The amount of wastewater generated by the oil-related and dyeing industries has increased annually. In the textile industry considered as one of the most polluting industrial sectors [1], water is mainly used for dyeing and finishing processes. Dyes including anionic and cationic dyes are kinds of textile dyes. Cationic dyes, called alkaline dyes, are soluble in water and ionize in aqueous solution to produce dyes with positively charged, colored ions. By contrast, anionic dyes, known as acidic dyes, dissolve in water in an anionic state. Wastewater containing oil-in-water (O/W) emulsions originates from petroleum-related industries, such as the petrochemical, leather, steel and metal processing industries. Its emissions often lead to specific and serious environmental issues [2,3,4,5]. Wastewater is one of the most intractable pollution sources affecting the natural environment, threatening the aquatic ecosystem and endangering human health in various ways. Many efforts have been made to remove dissolved organic dyes and O/W emulsions from the wastewater, in addition to traditional wastewater treatment methods including physical methods, chemical methods, and biological methods, as well as membrane filtration that has emerged in recent years. A membrane separation process has been applied in the fields of drinking water treatment, industrial wastewater treatment, and advanced treatment of wastewater because of its unparalleled advantages such as low energy consumption, no chemical reaction, high efficiency, simplicity and no pollution to the environment [6]. Water pollution has led to an emphasis on wastewater treatment. Various materials have been developed for removing contaminants in water. However, materials with high flux and rejection which can simultaneously separate and remove various wastewater impurities are rare.

Special functional materials with special wettability are taken into consideration for effective sewage separation. The hydrophilicity mainly depends on the performance of the functional groups on surface materials, and the construction of a hierarchical nanostructure hydrophilic surface can further lead to underwater superoleophobicity. In recent years, the application of anionic clay [7] has received increasing attention from the scientific community. Sep is an inexpensive anionic clay mineral with a rod-like microstructure. It is characterized by a very large surface area (>300 m^2^/g) in a relatively small porous volume (~0.4 cm^3^/g) [8,9]. Due to its porous structure, adsorption cations and large specific surface area, it can be used as an adsorbent to remove contaminants from solution [10]. Moreover, Sep has a large hydrophilic group on its surface that can be used to fabricate a hydrophilic membrane surface. However, Sep has the disadvantages of being brittle is easy to break and can fall off a film.

Hydrotalcite materials belong to layered compounds, which are easy to synthesize, non-toxic, inexpensive and exhibit a significant range of physical and chemical properties. The layered compound refers to a class of compounds having a lamellar structure, interlamellar ion exchangeability and layer cationic characteristics. MgAl-layered-double-hydroxide (MgAl-LDH) are synthetic solids with positively charged brucite-like layers of mixed metal hydroxides separated by interlayered hydrated anions, defined by the general formula: [M_1_^II^_−x_M^III^_x_(OH)_2_]^x+^[(A^n−^)_x/n_.yH_2_O]^x−^ [11]. Indeed, these properties make the MgAl-LDH excellent matrices of reception of numerous molecules. Therefore, the addition of Sep in the LDH layer can significantly improve the mechanical properties and separation performance of the membrane, not only can simultaneously adjust the water flux and wettability of the membrane, but also effectively remove harmful substances in the wastewater.

In this study, hierarchical structures inspired by natural antifouling surfaces (fish scales or underwater biological skins) have been proposed and designed for efficient pollution separation [12,13,14]. We designed and manufactured a robust three-dimensional (3D) hierarchically rough sheetlike membrane, namely MgAl-LDH and Sep, that were evenly covered over CA membranes through layers and layers, which removed contaminants through a simple and effective process. As shown in Figure 1, the 3D hierarcheical structure uses the CA film as a substrate, LDH is selected as a 2D layer, and then Sep was introduced as a 3D layer. The membrane surface morphology and surface roughness indicated that the membranes with a hierarchical structure were completely prepared [15]. The 3D hierarchical MgAl-LDH/Sep membranes exhibited a series of excellent performance for separation of oil and organic contaminants from the wastewater, and membranes could remarkably improve the rejection rate of dye and flux, and avoided membrane fouling in the oil-removing process [16,17,18,19,20,21,22]. This method has the advantages of simple operation, low cost, and environmental friendliness. It can provide a facile and environmentally friendly way for the fabrication of functional materials with potential for separation [23].

## 2. Experimental Section

### 2.1. Material

Magnesium nitrate hexahydrate (Mg(NO_3_)_2_·6H_2_O, ≥99%), aluminum nitrate nonahydrate (Al(NO_3_)_3_·9H_2_O, ≥99%), sodium nitrate (NaNO_3_), nitric acid (HNO_3_), sodium hydroxide (NaOH), sodium dodecyl sulfate (SDS) and cellulose acetate (CA) microfiltration membrane were purchased from Kelong chemical Co. Ltd., Chengdu China. CA microfiltration membrane with pore size of 0.22 μm (22 mm in diameter) was used as the substrate supporting the layer for modification. All other reagents were analytical grade and used without purification. Sepiolite was purchased from Sigma-Aldrich Co. Ltd. (Shanghai, China), and purified by NaCl solution. Ultrapure water was used throughout which obtained from a Milli-pore Mili-Q system (Boston, MA, USA).

### 2.2. Synthesis of MgAl-Layered-Double-Hydroxide (MgAl-LDH)

The MgAl-LDH in a molar ratio of M^2+^: M^3+^ (Mg^2+^: Al^3+^ = 4) was prepared by a traditional co-precipitation method [24]. Briefly, the metal nitrate mixture which contained of Mg(NO_3_)_2_•6H_2_O, Al(NO_3_)_3_•9H_2_O and NaNO_3_ was dissolved in 100 mL ultrapure water. Then, under constant agitation, the pH of the reaction system was controlled at 10 ± 0.5 by adding NaOH solution. The precipitates were stirred for another 30 min to obtain uniform mixture. Subsequently, the white precipitates were poured into thermal reactor (200 mL capacity) and kept 110 °C for 10 h. The mixtures were washed with deionized water and ethanol respectively and centrifuged until the pH approached neutral. The solid was obtained after drying at 60 °C for 12 h.

### 2.3. Fabrication of the MgAl-Layered-Double-Hydroxide/Sepiolite (MgAl-LDH/Sep) Membranes

MgAl-LDH/Sep membranes were fabricated via 3D hierarchical architectures construction methods, namely a method of assembling LDH and Sep onto the surface of the CA membrane through layer-by-layer sequentially at a suction vacuum pressure of 0.1 MPa. CA membrane was used as the supporting substrate. The vacuum filtration equipment is given in Appendix A. In detail, deionized water was added to the surface of the CA membrane to make sure the surface of the membrane was fully wetted. Then, LDH was deposited on the CA membrane. After this procedure, 0.002M Al^3+^ used to provide oppositely charged cations between LDH and Sep was added to the surface of LDH layer. Subsequently, the Sep was deposited on the LDH layer tightly due to hydrogen bonding and electrostatic attraction. The membrane was dried at room temperature. More details are given in the Appendix A. Membranes prepared with different Sep and LDH contents hereinafter are referred to as LDH-M, Sep-M, M1, M2, M3 and M4 respectively. More details and membrane compositions (Appendix A) are given in the Appendix A.

### 2.4. Preparation of Oil-in-Water (O/W) Emulsions

In a typical solution process, all kinds of O/W emulsion, SDS/ mesitylene /H_2_O emulsion, SDS/petroleum ether/H_2_O emulsion, SDS/decane/H_2_O emulsion and SDS/kerosene/H_2_O emulsion, were prepared by mixing the oil and water with a volume of 1:100, and the surfactant (SDS) concentration was 0.2 mg/mL. To be full emulsified, the mixture was ultrasonic-treated for 0.5 h and stirring for 3 h at room temperature. After emulsification, the redundant floating oil was removed. Finally, all the above emulsions were uniform for hours in laboratory environment.

### 2.5. Characterization

The MgAl-LDH/Sep composite membranes were analyzed for theirs morphology using scanning electron microscopy (SEM, JSM-7500F, JEOL, Tokyo, Japan). The phase structure and crystals orientation was identified by X-ray diffraction (XRD, PANalytical X’Pert Pro diffractometer, The Netherlands) with Cu Kα radiation source at a scan rate of 2° per min ranging from 5° to 70°. The surface morphology and surface roughness of the membrane were characterized by atomic force microscopy (AFM). The film surface was imaged with a scan size of 10 μm × 10 μm. Membrane surface chemical structure and composition were analyzed using Fourier transform infrared spectroscopy (FT-IR, Nicolet Avatar 370 FTIR spectrometer, Madison, WI, USA). The concentrations of dyes were determined spectrophotometrically by ultraviolet–visible (UV–Vis) spectrometry on a UV-762 (Shanghai precision scientific instrument co., Shanghai, China). The water contact angle and underwater oil contact angle were measured using a contact angle meter (Beijing Hake, XED-SPJ, Beijing, China). The MoticBA300Pol microscope (Nikon Digital Sight DS-F11, Tokyo, Japan) was employed to analyze the emulsions before and after filtering separation. Total organic carbon (TOC) was examined using a TOC meter (Shimadzu TOC-VCPH analyzer, Tokyo, Japan). Pore size properties of the membrane were measured by the method of bubble point and mean flow pore test, using the BSD-PB Membrane Pore Size Analyzer (Beishide Instrument 3H-2000PB, Beijing, China).

### 2.6. Permeation and Separation Performances

The permeation performances were implemented by a vacuum filtration experiment. The flux (J) of the membrane was obtained by permeating the solution through the membrane under a pressure of 0.1 MPa (1bar) and was calculated by Equation (1).

The rejection rates (R) for various organics and heavy metal ions were performed through a filtration system with an effective membrane area of 12.566 cm^2^ to evaluate the separation performances of the membranes. In detail, 20 mL of synthetic organics solution was filtered across the membrane. The solution concentration was determined by using an ultraviolet–visible (UV–Vis) spectrophotometer and TOC. The rejection rate R was calculated according to the following Equation (2):(1)$$J=\frac{V}{A\times t}$$
(2)$$R=\left(1-\frac{C_{p}}{C_{f}}\right)\times 100\%$$
where *V* (L) is the permeated volume of the solution; *A* (m^2^) is the effective membrane area, *t* (h) is the testing time. R is the rejection rate of organics and heavy metal ions. *C_f_* (mg-L^−1^) is the concentration of the original solution and *C_p_* (mg-L^−1^) is the concentration of the filtered solution, respectively. More details are given in the Appendix A.

## 3. Results and Discussion

### 3.1. Characterization

#### 3.1.1. X-ray Diffraction (XRD)

X-ray diffraction was performed to characterize the crystalline structures of the materials. The XRD patterns of Na^+^-Sep and MgAl-LDH(-NO_3_^−^) have been depicted in Figure 2a. The diffraction peaks of MgAl-LDH(-NO_3_^−^) were typical of the layered double hydroxides structure with sharp and symmetric reflections of the basal (003), (006) and (009) planes, and broad, less intense and asymmetric reflections for the nonbasal (015) and (018) planes [25]. The XRD pattern in Figure 2a could be considered as possessing a layered structure. In addition, the intensity peak was obtained at 2θ = 7.48° (110) with a d-spacing of 1.08 nm, which confirmed the structure of the Sep.

#### 3.1.2. Fourier Transform Infrared Spectroscopy (FT-IR)

The FT-IR spectra of Na^+^-Sep and MgAl-LDH(-NO_3_^−^) are shown in Figure 2b. In the case of Na^+^-Sep, the characteristic peak observed at 3686 cm^−1^ was attributed to triple bridge group Mg_3_OH. The stretching vibration and the OH-bending mode at 3425 cm^−1^ were related to zeolitic water [26]. The silicate was represented by band in the range of 1200–400 cm^−1^. The features located at 1612 and 1402 cm^−1^ could be assigned to Si–O–Si bonds and bands at 1025 and 465 cm^−1^ correspond to the stretching vibration of Si–O. For MgAl-LDH(-NO_3_^−^) composite, the broad band at 3450 and 1627 cm^−1^ were due to the stretching and bending vibrations of O–H in the LDH crystalline structure. The stretching vibration of NO^3−^ was responsible for the sharp band at 1384 cm^−1^. The absorption band at around the 400-800 cm^−1^ region could be ascribed to the lattice vibration of Mg^2+^–O, Al^3+^–O, O–Mg^2+^–O and O–Al^3+^–O groups [27].

A membrane was prepared with different contents including Sep, LDH and MgAl-LDH/Sep (1:2) hereinafter referred to as Sep-M, LDH-M and M2 respectively. It can be seen that the formed composite films did not fall off easily and were and relatively stable. Even if the membrane was bent with tweezers, there was no problem such as breakage or shedding. In contrast, the rod-like structure of pure Sep results in imperfect coverage of the membrane surface which can easily fall off and snap.

#### 3.1.3. Scanning Electron Microscopy (SEM)

The surface and cross-section microstructure morphology of the membranes was observed by scanning electron microscopy (SEM) and the results are presented in Figure 3 and Figure 4. In Figure 3a, the pure LDH membrane exhibited a uniform platelet-like morphology. Along with the hydrogen bond interaction, LDH was coated on the CA membrane surface tightly with single LDH nanosheets packed together. After modification, the Sep with rod-like structure covered the LDH membrane surface, thereby forming a rough micro-/nano-scale hierarchical structure (Figure 3b,c). The SEM image shows that the texture of these Sep piles resemble weeds. Because of the strong electrostatic interaction and hydrogen bond between LDH and Sep, and the loose LDH, in some areas, LDH broke away from the original layer and inserted into the layer of the Sep. With the increase of Sep, the Sep uniformly covered the surface of the membrane which is displayed in Appendix A. In addition, the composite membrane has also exhibited a porous structure and obvious nano-particles on the membrane surface as indicated by the red circles in the figure [28].

Furthermore, the SEM image of the membrane cross section is shown in Figure 4d–f. In Figure 4d,d1, the cross section of the LDH membrane was compactly packed with a large amount of LDH sheets, the thickness was about 110 μm. Figure 4e,f shows that the addition of Sep significantly altered the cross-section microstructure of these prepared membranes. After Sep modification, the membrane surface exhibited a 3D hierarchical structure, which indicated the successful formation of the MgAl-LDH/Sep membrane. As Figure 4e,e1 shows, the membrane was divided into three layers as shown by the red circle in the cross section of the membrane, and from top to bottom were a Sep layer, an LDH layer and a CA membrane layer. Many micron-sized pores were distributed, forming a microscale and nanoscale hierarchical composite structure. These results demonstrated that the composite membranes with the desired microstructure have been successfully prepared. The SEM images for M3 and M4 membrane cross-sections are shown in Appendix A.

#### 3.1.4. Atomic Force Microscopy (AFM)

The change in surface roughness of LDH-M and M2 membranes was observed through Atomic force microscopy (AFM). Two- and three-dimensional AFM images are shown in Figure 5. Moreover, roughness parameters such as Ra and Rq of composite membranes were shown and calculated by using the AFM analysis software based on an AFM scanning area of 10 μm × 10 μm. As displayed in Figure 5, the bright and dark regions represent the highest and lowest points of the membrane surface, respectively. The valleys or the pores of the membrane in the 3D images can be seen on the surface of the composite membrane. The obtained surface roughness parameters are listed in Appendix A. The results show the average roughness (Ra) values of LDH membrane is 72.1 nm, with the corresponding values of Rq being 106 nm. With the increase of Sep, the roughness of M2 composite membrane gradually increases, and some large “peak” appears on the membrane surface, as shown in Figure 5b. Obviously, the roughness of M2 composite membrane is significantly higher than that of LDH-M membrane. The results can be explained by the fact that the addition of the Sep leads to an increase in the average roughness, which can be attributed to the average pores diameter and surface porosity alteration caused by the accumulation of nanomaterials on the membrane surface. Roughness enhancement usually causes two major changes in membranes including an increase in effective filtration area, resulting in an increase in membrane flux and a decrease in antifouling performance [29]. However, the antifouling property is determined by the hydrophilicity of the membrane, contact angle, pore diameter and membrane structure. 2D AFM images also evidence that the Sep and LDH have covered the surface and constituted the upper surface layer as the arrows shown. Pore size distribution of the membrane is one of the crucial factors considered to evaluate the membrane performance. Figure 5e,f illustrates the pore size distribution of LDH-M and M2 membranes. It can be seen that, the LDH-M membrane has a bigger average pore size than the M2 membrane. The obtained results showed that the average pore size of M2 membrane was 0.0377 μm according to Appendix A. Accumulation of the composite on the surface of the membrane by the layer-by-layer method could be the main reason for the reduction of average pore size. The reduction in the average pore size could increase the rejection rate of the membrane. Moreover, due to the pore size decreasing, the flux of the membrane also declines slightly. Nevertheless, the membrane flux could be related to the hydrophilic performance, construction method, pore size and other factors.

### 3.3. Wettability of MgAl-LDH/Sep Membrane

It was anticipated that the surface wetting behavior of the membrane was determined by the chemical composition and structure of its surface [30]. Owing to the abundant hydrophilic groups of Sep (Figure 2b), an anionic clay, the membrane exhibited the desired hydrophilicity, which was evaluated by the water contact angles in air and the underwater oil contact angle. As shown in Figure 6c, the water contact angle of the M2 membrane was about 8 and after a few seconds, the water contact angle was close to about 0, which could be attributed to the synergistic effect between the hydrophilicity of the Sep and the hierarchical structure, and the tremendous hydroxyl groups on the Sep could enhance hydrophilicity and the interaction between the membrane surface and water. In contrast, the CA-M and the LDH-M exhibited higher water contact angles of 53.33° and 24.17°, revealing that the addition of Sep to the membrane could increase hydrophilicity. Furthermore, the membrane could not be contaminated with various oils, including mesitylene, petroleum ether, kerosene and decane, confirming the underwater oleophobicity of the membrane. The underwater with oil contact angles of 155.38° for mesitylene 156.83° for petroleum ether, 156.99° for kerosene and 157.13° for hexane are shown in Appendix A.

Moreover, the MgAl-LDH/Sep membranes were low adhesiveness to underwater oil droplets (Figure 7). For high oil adhesion materials, the deformation of oil droplet usually appeared because the adhesion force between the oil droplet and the surface of membrane produced vertical tensile stress when the oil droplet was separated [11]. Therefore, the underwater low oil adhesion behavior of MgAl-LDH/Sep membrane was attributed to the membrane surface possessing a large amount of hydroxyl groups and high surface energy, indicating that the hydrophilicity of the membrane would facilitate the penetration of the water through the membrane in the separation process. These advantages indicated that the membrane had superhydrophilicity and low underwater oil adhesion, which would be an ideal membrane material for separating organics.

### 3.4. Dye Removal Separation Performance

Separation performance of the obtained membranes (LDH-M, M1, M2, M3, M4) for dye wastewater was investigated through a filtration test. Four anionic and cationic dyes, Congo red (CR), Acid red 87 (AR), Methylene blue (MB) and Methyl orange (MO) were chosen to evaluate dye rejection of the membranes. Herein, only experimental data for MB and CR, as anionic and cationic dyes respectively, were listed to illustrate separation performance. The other results are shown in Appendix A. The fluxes of the composite membranes to different ionic dye solutions are depicted in Figure 8a,d. All the tested dye solutions fluxes showed a downward trend but still maintained a high value of about 900 L·m^−2^·h^−1^. The decline of flux was mainly due to the accumulation of dye molecules on membrane surface and the increase in membrane thickness. The removal efficiency of the composite membranes is illustrated in Figure 8b,e. In this image, it was concluded that, for MB (via LDH-M), the rejection rate was greatly reduced from 52.70% to 45.02% as the dye concentration increased from 10 to 30 mg/L. For CR (via LDH-M), from 100 mg/L to 300 mg/L, the rejection decreased from 99.00% to 95.70% However, when the as-prepared M2 was used to dispose of dyes MB and CR, the total rejection rates were about 99.90% and 99.80%, respectively. The decolorization effect was visually confirmed by photographs of samples of the stock and the filtered dye aqueous solutions (via LDH-M, M1, M2) as shown in Figure 8 inset (1), (2). The concentration of dyes and the filtered solutions measured by UV–Vis are shown in Figure 8c,f. And the original solution of MB was diluted by one time to ensure the accuracy of the data. The other UV–Visible absorption spectra are shown in Appendix A. The obtained results revealed that the composite membrane had excellent separation performance for dyes compared with other membranes listed in Table 1 [31,32,33,34,35].

As stated in the literature, the as-prepared membranes were used to remove the ionic organic dyes from aqueous solution and to explore the possible mechanism of the interaction between the membrane surface and dye molecules [36]. B. Vander Bruggen et al. believed that due to the stronger effects of steric hindrance and electrostatic repulsion, organic dyes with more positive or higher molecular weight were generally more effectively repelled by positively charged membranes [36]. The as-prepared composite membrane herein was negatively charged, which resulted in an effective electrostatic interface between the dye molecules and the membrane when the dye molecules exhibited a positive charge. The high rejection rate of the negative charged dye CR was mainly due to its high molecular weight.

### 3.5. Oil/Water Separation Performance

It was well-known that the excellent superhydrophilicity and oleophobicity of the MgAl-LDH/Sep membrane would also enhance the membrane’s ability to separate O/W emulsions. Mesitylen/water emulsion was used as an example to demonstrate the oil-water separation performance of the composite membrane. The photographs and microscope images of the stock solution and filtered emulsion were shown in Figure 9. The oil/water emulsions were milky white liquids (Figure 9b) before separation. There were many oil droplets (Figure 9c) with a size about 1–14 μm in the as-prepared SDS/mesitylen/H_2_O emulsion. After separation, the collected filtrates (Figure 9b) were transparent and there were no observable oil droplets in the filtrate, indicating the successful separation of the oil in emulsion.

The O/W separation performance of as-prepared membranes was evaluated by pouring the O/W emulsion on to the membrane and separating it under vacuum filtration at a pressure of 0.1 MPa. The separation flux and oil rejection rate of the membrane in separation process was investigated to evaluate the separation ability. As shown in Figure 6, the oil flux values of membranes were 301.6, 318.3, 106, 222.9 L·m^−2^·h^−1^ for mesitylene, petroleum ether, kerosene and decane. In addition, the separation efficiency of the membrane for different types of oil/water mixtures was above 90%. The results demonstrated that the composite membrane has excellent oil-water separation performance.

### 3.6. Durability and Stability of MgAl-LDH/Sep Membranes

The durability and stability of the obtained membrane were considered to be important criterions for practical oil-water separation applications. To verify the durability and stability of the filtration membrane, mesitylene was taken as an example to evaluate the emulsion separation ability of the membrane by cyclic experiments. The separation flux and oil rejection rate of the membrane of O/W emulsion in 9 cycles are displayed in Figure 10. It can be seen that the separation flux decreases slightly with the number of cycles. Surfactants tend to adsorb on the membrane surface and alter membrane wetting properties during oil/water separation, which can significantly affect separation performance [37]. This result corresponded to the change in the contact angle of the submerged oil in Figure 10.

Moreover, the stability of the superhydrophobicity and superoleophilicity membrane was investigated in harsh conditions, such as acidic, alkaline and salty environments. It was well known that under extreme conditions, the surface wettability of the membrane changed. However, as given in Figure 11, the membranes were still superoleophobic which was confirmed by the underwater mesitylene contact angle of 156.59°, 156.20° and 155.98° in the aqueous solutions of HCl (pH = 1), NaCl (pH = 7) and NaOH (pH = 13) respectively, indicating that the membrane could withstand acidic, alkaline and salty environments. Furthermore, the stability of the membrane in water was another important limitation for the durability. In Appendix A, the membrane was placed in the water for 30 days without any change in the as-prepared membrane surface. Therefore, the membrane with prominent stability and durability could be used for long-term wastewater separation.

### 3.7. Other Water Treatment Applications

The MgAl-LDH/Sep membrane could be applied to multifunctional water treatment. For example, besides methylene blue (MB) and oil-water (O/W), some toxic heavy metal ions could also be partly removed by the MgAl-LDH/Sep membrane in the wastewater contaminants removal process. Figure 12. shows the removal rates of four ions of Fe^3+^, Pb^2+^, Cu^2+^ and Ni^2+^. The rejection rates of Fe^3+^, Pb^2+^, Cu^2+^ and Ni^2+^ were 90.33% ± 0.15%, 83.33 ± 0.15%, 56.00% ± 0.17% and 45.67% ± 0.12%, respectively. These results indicated that the as-prepared membrane could simultaneously reduce the concentration of various pollutants in wastewater.

## 4. Conclusions

In this study, the novel MgAl-LDH/Sep membranes were prepared by layer by layer processing on a cellulose acetate membrane, which could simultaneously remove anionic and cationic dyes, oil/water emulsions and heavy ions. It was found that the resulting composite membranes exhibited high water flux (about 1200 L·m^−2^·h^−1^) and could be applied to separate various anionic and cationic dyes in terms of their rejection efficiencies for MB and CR (above 99.8%) and their flux (about 900 L·m^−2^·h^−1^). Moreover, MgAl-LDH/Sep composite membranes had outstanding hydrophilicity and underwater superoleophobicity, achieving better oil repellent performance and recycling property, which could maintain steady flux and retention efficiency after nine cycles. Meanwhile, the composite membranes exhibited excellent stability under harsh environments. The characteristics of the as-prepared membrane were advantageous for the development of membrane materials for sewage treatment.

## Figures and Tables

**Figure 1 polymers-11-00525-f001:**
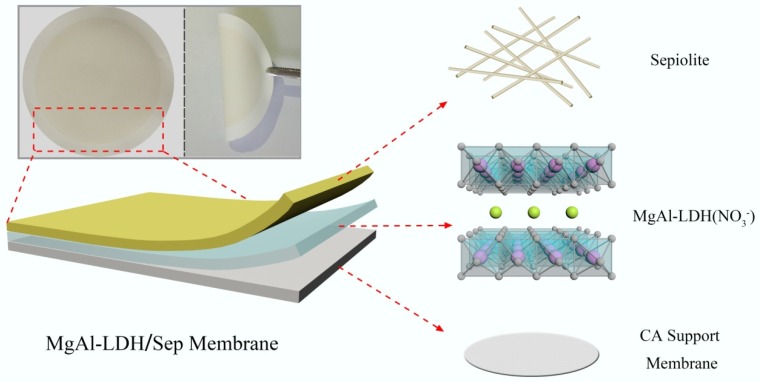
Schematic illustration of 3D hierarchical architecture.

**Figure 2 polymers-11-00525-f002:**
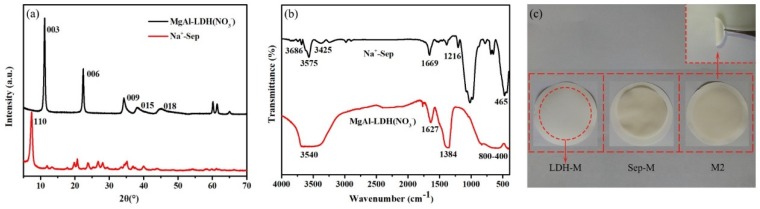
Characterization of structural properties of the prepared materials (**a**) X-ray diffraction (XRD) patterns. (**b**) Fourier transform infrared (FT-IR) spectra. (**c**) The photograph of the LDH-M, Sep-M and M2.

**Figure 3 polymers-11-00525-f003:**
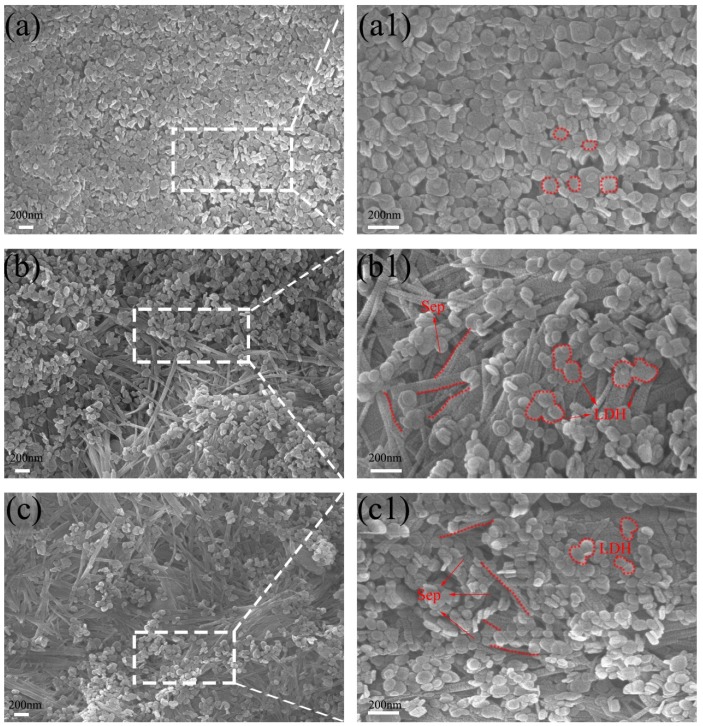
Scanning electron micoscope (SEM) image for LDH-M (**a**), (**a1**), M1 (**b**), (**b1**) and M2 (**c**), (**c1**) of composite membrane surface.

**Figure 4 polymers-11-00525-f004:**
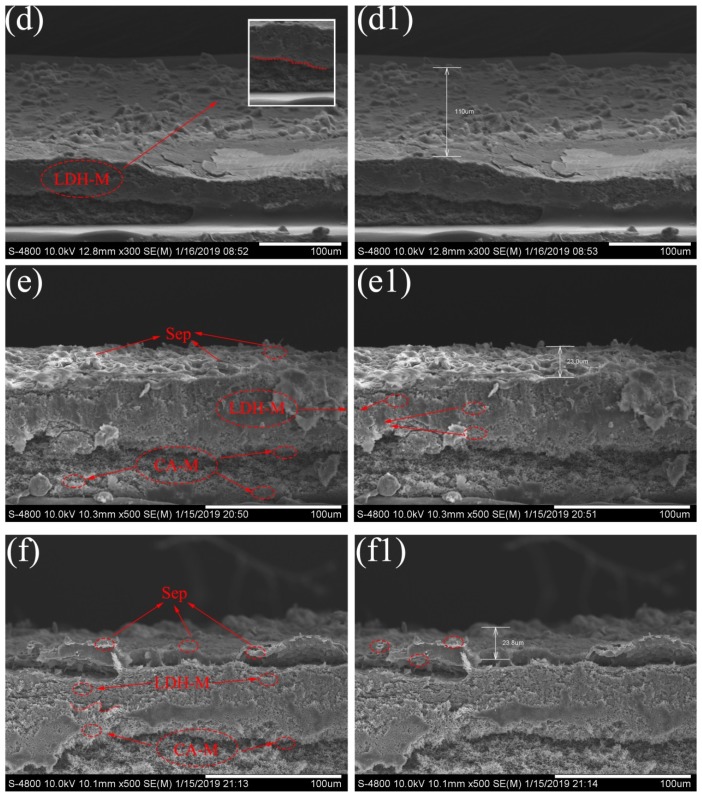
SEM image for LDH-M (**d**), (**d1**), M1 (**e**), (**e1**) and M2 (**f**), (**f1**) of membrane cross-section.

**Figure 5 polymers-11-00525-f005:**
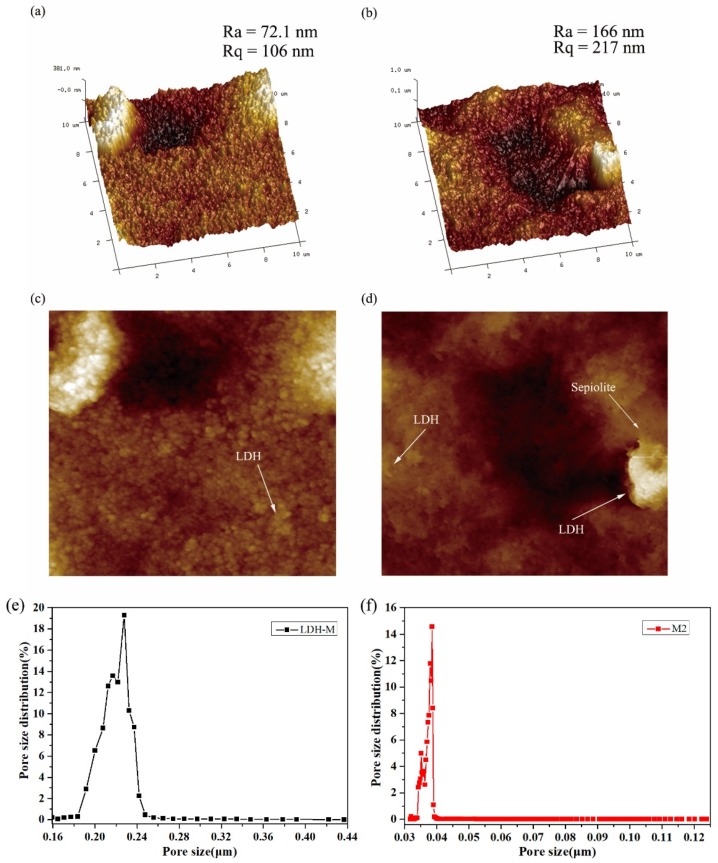
Atomic force microscope (AFM) and pore size distribution images of prepared membrane. (**a**,**c**) AFM images of LDH-M; (**b**,**d**) AFM images of M2. (**e**) Pore size distribution of LDH-M; (**f**) Pore size distribution of M2.

**Figure 6 polymers-11-00525-f006:**
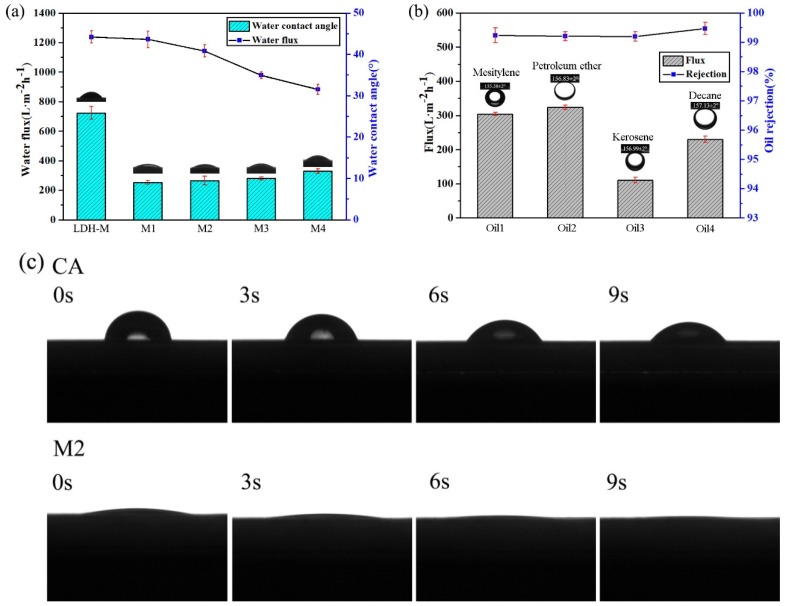
Characterization of the wettability of MgAl-LDH/Sep membrane (**a**) water contact angle and water flux (**b**) oil rejection rate and flux (**c**) water contact angle changes.

**Figure 7 polymers-11-00525-f007:**
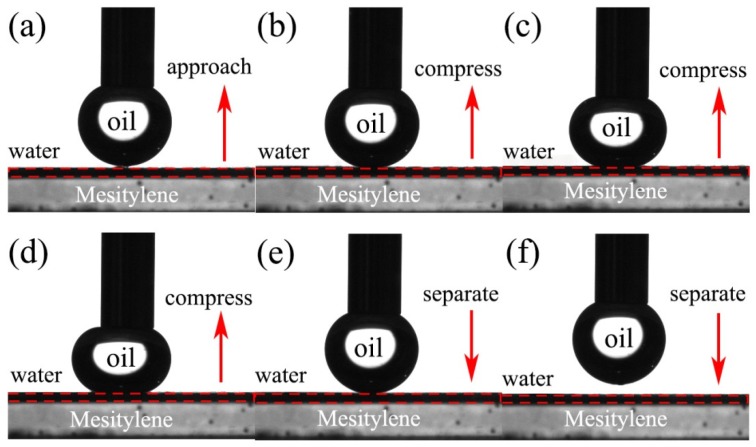
Dynamic approach-compress-separate mesitylene adhesion test of MgAl-LDH/Sep membrane. (**a**) Oil droplets approach the membrane surface; (**b**–**d**) Oil droplets compress the membrane surface; (**e**) and (**f**) Oil droplets separate the membrane surface;

**Figure 8 polymers-11-00525-f008:**
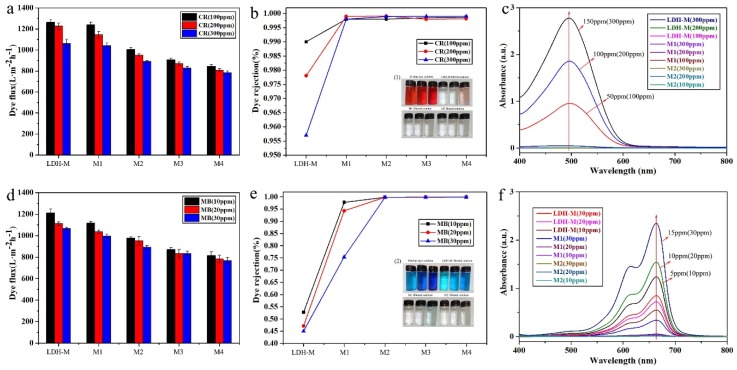
Results for dyes separation. (**a**), (**d**) dye flux (**b**), (**e**) dye rejection (**c**), (**f**) UV–Visible absorption spectra.

**Figure 9 polymers-11-00525-f009:**
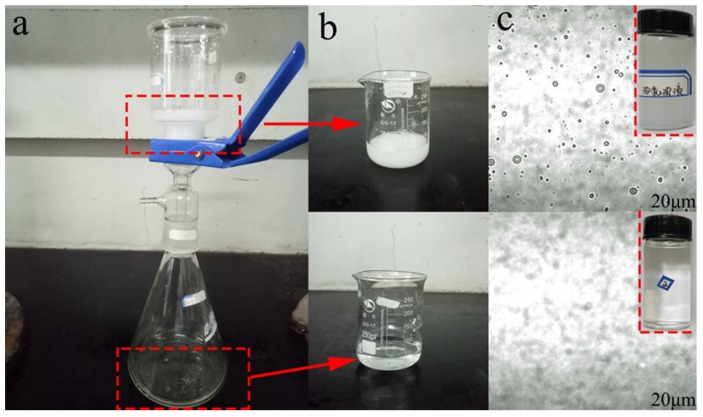
(**a**) the photograph of oil filtration apparatus (**b**) oil-in-water (O/W) emulsion and the filtered emulsion (**c**) the optical microscope images.

**Figure 10 polymers-11-00525-f010:**
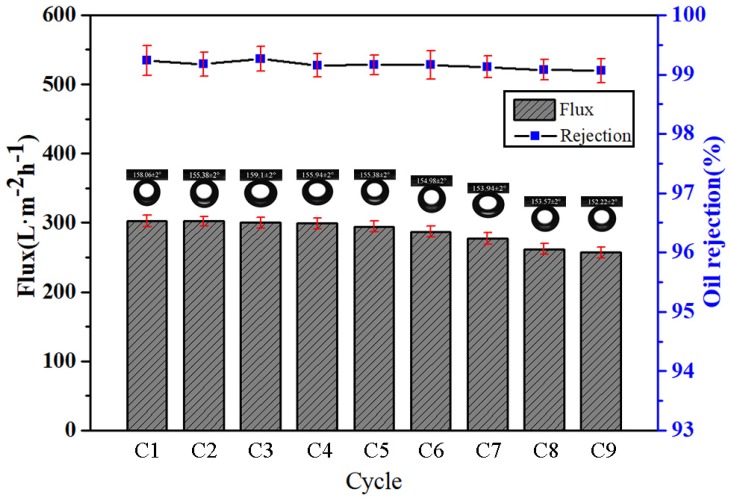
Characterization of the O/W separation performance of the prepared membranes.

**Figure 11 polymers-11-00525-f011:**
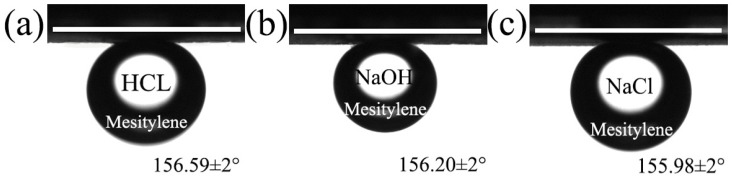
The underwater oil contact angle of mesitylene on the MgAl-LDH/Sep membrane under HCl (pH = 1), NaCl (pH = 7) and NaOH (pH = 14) solution. (**a**) HCl (pH = 1); (**b**) NaOH (pH = 14) (**c**) NaCl (pH = 7).

**Figure 12 polymers-11-00525-f012:**
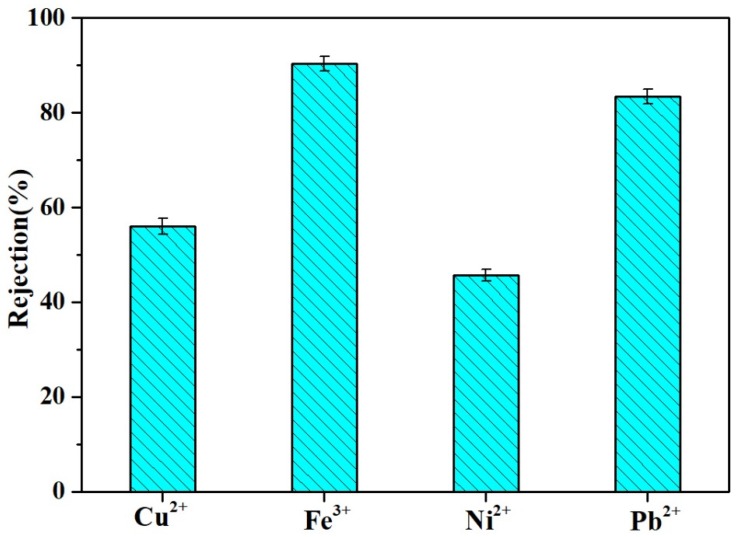
Heavy metal ions rejection.

**Table 1 polymers-11-00525-t001:** Comparisons of the performance of MgAl-LDH/Sep membrane with that of commercially available and literature reported dye separation membranes.

Membrane	Pure Water Flux (L·m^−2^·h^−1^·bar^−1^)	Dye Permeate Flux (L·m^−2^·h^−1^·bar^−1^)	Dye Rejection (%)	Operating Conditions		Ref.
				pH limitation	Max. Temp. (°C)	
UH004 (hydrophilic PES)	27.5	27	98.9	0–14.0	95	[31]
SIO2-PIL/PES blending NF	21.2	null	90-93	–	–	[32]
GO-PSBMA/PES blending NF	12.5	8.8	95	–	–	[33]
Zwitterion-hydrotalcite incorporated PES	20.1	null	86.7	–	–	[34]
PAEK-COOH	29.5	25	95	1.0–10.0	95	[35]
MgAl-LDH/Sep	1200	900	99.8	–	–	This work

Operation pressure: 0.4 MPa. This work: 0.1 MPa.

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
