# Peer review of "MgAl-Layered-Double-Hydroxide/Sepiolite Composite Membrane for High-Performance Water Treatment Based on Layer-by-Layer Hierarchical Architectures"

_polymers, 2019, doi:10.3390/polym11030525_

Reviewer 1 Report

Authors studied LBL membranes for water treatment. Article is not written properly (scientific way). Some are mentioned below.

Line 19:  Membrane materials or membrane?

Line 23:  Basment of membrane?   Is it supporting layer?

Line 72-77: This part should be either in Experimental or in Discussion,

Line 96-94:  2.2. Synthesis of MgAl-LDH :  Please give reference.

Line 106;  MgAl-LDH/Sep membranes were fabricated via in 3D hierarchical architectures construction   

          Methods…  Is there any reference?

Characterization of membrane should be given in section,

i)                   FTIR

ii)                SEM

iii)             AFM

iv)              Contact angle

Line164-165: ; “stretching vibration and the OH-bending mode at 3425 cm-1 were related to zeolitic water”.  Please give reference.

Line 173; “The photograph of the membrane was displayed in Fig.1c”.  It is not necessary. and should be remove.

Line 209: From AFM images authors should calculate pore sizes and pore sizes distribution curve and discuss their findings with membrane performance.

Line 247-250:   Fig. 6 is not necessary. Results are mentioned in the text. It should be deleted.

Line 265 Fig 9. The photograph of membrane. (a) M2 (b) MB filtered M2 265.  It should be deleted.

Line 325: Fig 12 shoud be deleted.  It is discussed in the text

This article should be accepted for publication after measure revision

Author Response

Dear editor and reviewers:

Thank you for taking time out of your busy schedule to review our manuscript (polymers-458077). Your comments are really thoughtful and helpful, which would help us in depth to improve the quality of the paper. Based on these comments and suggestions, we have made careful modifications on the original manuscript. Here we submit our revised manuscript with the title “MgAl-Layered-Double-Hydroxide/sepiolite composite membrane for high-performance water treatment based on layer-by-layer hierarchical architectures”. And revised portion are marked in red in the article and all the changes were illuminated in this text. We hope the new manuscript will meet your magazine’s standard. If you and reviewers have any further questions, please do not hesitate to contact us.

Sincerely yours

Zongxue Yu

Response to Reviewer 1 Comments

Point 1:Line 19:  Membrane materials or membrane?

Response 1: Thank you for your reminder. We are very sorry for our negligence. The sentence revised as below:

These membranes were analyzed by electron microscopy (SEM), atomic force microscope (AFM) and X-ray diffraction (XRD), etc.

Point 2:Line 23:  Basement of membrane?   Is it supporting layer?

Response 2: Thank you. Yes, the basement membrane refers to the supporting layer, which is the CA membrane in the paper. In order to avoid unnecessary misunderstanding, we have revised the basement membrane to CA membrane.

Point 3:Line 72-77: This part should be either in Experimental or in Discussion.

Response 3: Thank you for your comments. Considering the Reviewer’s suggestion, we have made corrections in the text. We further discuss this part in the discussion and experimental section. We added relevant content in the section of the fabrication of the MgAl-LDH/Sep membranes and the discussion of the SEM cross section images. The changes are marked in red.

Point 4:Line 96-94:  2.2. Synthesis of MgAl-LDH: Please give reference.

Response 4: Thank you for your suggestion. I have inserted relevant references in the text and have readjusted the order of the references.

The literature is listed below:

[1] Wang J , Wang X , Tan L , et al. Performances and mechanisms of Mg/Al and Ca/Al layered double hydroxides for graphene oxide removal from aqueous solution[J]. Chemical Engineering Journal, 2016, 297:106-115.

Point 5:Line 106:  MgAl-LDH/Sep membranes were fabricated via in 3D hierarchical architectures construction.    Methods…  Is there any reference?

Response 5: Thank you for your professional comment. In this paper, MgAl-LDH/Sep membranes were fabricated via 3D hierarchical architectures construction methods, namely a method of assembling LDH and Sep onto the surface of the CA membrane through layer-by-layer sequentially. This method is reported in some literatures. For example, Ding W et al. fabricated the organic-inorganic nanofiltration membrane using ordered stacking SiO2 thin film as rejection layer assisted with layer-by-layer method [1]. Weng G M et al. synthesized the cross-functional semi-transparent MXene-carbon nanotubes composite films through layer-by-layer assembling method [2]. Li F et al. modified cellulose acetate membranes with hierarchical structure [3]. Shami Z et al. successfully developed the 3D hierarchical roughness MgAl-layered double hydroxide branched to an electrospun porous nanomembrane for oil removing [4].

The literature is listed below:

[1] Ding W , et al. Fabrication of organic-inorganic nanofiltration membrane using ordered stacking SiO2 thin film as rejection layer assisted with layer-by-layer method. Chemical Engineering Journal, 2017. 330: p. 337-344.

[2] Weng G M , Li J , Alhabeb M , et al. Layer-by-Layer Assembly of Cross-Functional Semi-transparent MXene-Carbon Nanotubes Composite Films for Next-Generation Electromagnetic Interference Shielding[J]. Advanced Functional Materials, 2018.

[3] Li F , Gao R , Wu T , et al. Role of layered materials in emulsified oil/water separation and anti-fouling performance of modified cellulose acetate membranes with hierarchical structure[J]. Journal of Membrane Science, 2017, 543.

[4] Shami Z, Amininasab SM & Shakeri P 2016: Structure–Property Relationships of Nanosheeted 3D Hierarchical Roughness MgAl–Layered Double Hydroxide Branched to an Electrospun Porous Nanomembrane: A Superior Oil-Removing Nanofabric. ACS APPL MATER INTER 8, 28964-28973.

Point 6:Characterization of membrane should be given in section,

FTIR

SEM

AFM

Contact angle

Response 6: Thank you for your reminder. I have made a correction in the article, adding a subtitle to each part of the characterization. As follows:

XRD

FTIR

SEM

AFM

Since the contact angle is mainly used to characterize the wettability properties of the membrane and the headline is set here, the subtitle of the contact angle is not separately set here.

Point 7:Line164-165: “stretching vibration and the OH-bending mode at 3425 cm-1 were related to zeolitic water”.  Please give reference.

Response 7: Thank you for your valuable advice. Based on the revision, we have inserted relevant literature in the text.

The literature is listed as follows:

[1] Yu S , Zhai L , Zhong S , et al. Synthesis and structural characterization of magnetite/sepiolite composite and its sorptive properties for Co(II) and Cd(II)[J]. Journal of the Taiwan Institute of Chemical Engineers, 2016, 59:221-228.

Point 8:Line 173: “The photograph of the membrane was displayed in Fig.1c”.  It is not necessary and should be remove.

Response 8: Thank you for your scientific comments. We strongly agree with your opinion. This part has been deleted, and the relevant content has been modified accordingly.

Point 9:Line 209: From AFM images authors should calculate pore sizes and pore sizes distribution curve and discuss their findings with membrane performance.

Response 9: Thank you and we have attached great importance to this comment. For the characterization of the pore size and pore size distribution of the separation membrane, the most standard is to measure the pore throat pore size to characterize the narrowest and most effective pore size in this pore. However, AFM can only characterize the cortical pore size of the membrane surface and cannot effectively measure its internal effective pore size. So we used the bubble point-average flow method test instead of AFM to measure its pore size distribution. Test methods for pore size properties of membrane filters-Bubble point and mean flow pore test adopt the national standard of the People's Republic of China (GB/T 32361-2015). The results and discussion have been shown in the article.

Point 10:Line 247-250:   Fig. 6 is not necessary. Results are mentioned in the text. It should be deleted.

Response 10: Thanks for your advice. We agree with the comment and have made correction according to the Reviewer’s comments.

Point 11:Line 265: Fig 9. The photograph of membrane. (a) M2 (b) MB filtered M2 265.  It should be deleted.

Response 11: Thank you for your professional advice which made us realize a mistake. Figure 9 is not reflected in the article. We are sorry for our errors. This section has been removed and the order of the remaining diagrams has been changed.

Point 12:Line 325: Fig 12 should be deleted.  It is discussed in the text.

Response 12: Thank you for your thoughtful suggestions. After careful consideration, we have a little different idea. We believe that the graphical description is more intuitive to show whether the underwater oil contact angle of the membrane can maintain its original performance in a harsh environment. The combination of graph and discussion can better express the performance stability of the membrane in the acid-base salt environment. So we hope to keep Figure 12.

Reviewer 2 Report

In the manuscript polymers-458077 entitled “MgAl-Layered-Double-Hydroxide/sepiolite composite membrane for high-performance water treatment based on layer-by-layer hierarchical architectures”, the authors fabricated new MgAl-LDH/Sep composite membranes with hierarchical structure. It was reported that the prepared membranes showed superb permeation flux and rejection when used for filtration of dyes and oil/water emulsions.

The manuscript provided interesting results, however, it does not meet the requirements to be published in the journal of Polymers in the present form. The manuscript requires major revision by addressing the following comments:   

Scientific Questions:

1- The hydraulic permeability and molecular weight cut-off (MWCT) of the membranes need to reported. The authors are strongly recommended to follow the standard procedures in the literature to obtain these two important parameters.

2- The Figure 2(b1) shows that the LDH particles are deposited over the rod-like Sep layer. However, it was mentioned in section 2.2 that the Sep was deposited over the LDH layer. Explanation is needed here.

3- Figure 6: The variation of contact angle data can be due to variation of surface roughness and surface porosity of different samples. How did the authors consider these influential parameters?

4- It was claimed that the prepared composite membranes have high thermal stability, however, no experimental characterization is provided. How did the authors evaluate the thermal stability of the prepared membranes?

Non-scientific comments:

1- English (grammar, type, and punctuation) needs significant improvement. Few examples are:

-Page 1, Abstract, line 15: “is designing a highly efficient and flux material”

- Page 1, Abstract, line 18: “composite membranes hierarchical architecture via in 3D hierarchical architectures construction methods.”

- Page 10, line 252: “membranes also were low oil adhesiveness to underwater oil”

- Page 15, line 358: “membranes with the multifunction of anionic and cationic dyes”

2- Page 3, “Scheme 1” should be named as “Figure 1”. The number of other figures need to be changed accordingly.

3- Page 5, line 175: The stability of the composite film cannot be evaluated based on SEM photograph. Revision is needed.

4- Page 8, Figure 4, the range of the vertical axis of the AFM images needs to be equal to allow easier comparison of the surface topography.

5- Figure 11, the Y-axis for rejection should be changed to 93 to 100 instead of 0.93 to 1.0. Same comment for Figure 13.

6- Figure 12 & Figure S4: Numbers should be presented with similar significant digits.

7- Table S2: The surface roughness data is suggested to be presented in unit of "nm". Also, standard deviation needs to be reported for the data.

Author Response

Dear editor and reviewers:

Thank you for taking time out of your busy schedule to review our manuscript (polymers-458077). Your comments are really thoughtful and helpful, which would help us in depth to improve the quality of the paper. Based on these comments and suggestions, we have made careful modifications on the original manuscript. Here we submit our revised manuscript with the title “MgAl-Layered-Double-Hydroxide/sepiolite composite membrane for high-performance water treatment based on layer-by-layer hierarchical architectures”. And revised portion are marked in red in the article and all the changes were illuminated in this text. We hope the new manuscript will meet your magazine’s standard. If you and reviewers have any further questions, please do not hesitate to contact us.

Sincerely yours

Zongxue Yu

Response to Reviewer 2 Comments

Point 1:The hydraulic permeability and molecular weight cut-off (MWCT) of the membranes need to reported. The authors are strongly recommended to follow the standard procedures in the literature to obtain these two important parameters.

Response 1: Thank you and we have attached great importance to this comment. We are very sorry for our errors. We refer to some literatures about separation membranes for the calculation of flux. For separation membranes, the water flux was usually calculated by the following equation (1).

              (1)

Where V (L) is the permeated volume of the solution; A (m2) is the effective membrane area, t (h) is the testing time, and P (bar) is the applied pressure.

The literature is as follows:

[1] Chen T , Duan M , Shi P , et al. Ultrathin Nanoporous Membranes Derived from Protein-based Nanospheres for High-performance Smart Molecular Filtration[J]. J. Mater. Chem. A, 2017:10.1039.C7TA06800J.

[2] Chen L , Moon J H , Ma X , et al. High performance graphene oxide nanofiltration membrane prepared by electrospraying for wastewater purification[J]. Carbon, 2018:S000862231830071X.

[3] Soyekwo F , Zhang Q , Gao R , et al. Cellulose nanofiber intermediary to fabricate highly-permeable ultrathin nanofiltration membranes for fast water purification[J]. Journal of Membrane Science, 2017, 524:174-185.

Figure 1. Flux of M2 membrane

Figure 2. Pore size distribution images

Table 1. Pore size data

Name

LDH-M

M2

Measured   bubble point pressure (bar)

1.1026

3.8881

Measured   bubble point flow (L/min)

0.0066

0.0041

Minimum   pore size pressure (bar)

3.0256

15.135

Average   pore size pressure (bar)

2.155

12.7582

Gas   permeability (m3/(m2.pa.s))

3.94E-02

7.09E-03

Gas   flux (m3/(m2.h)) (ΔP=0.1000bar)

1.45E+02

1.20E+01

Measured   bubble point pore size (ÎĽm)

0.4367

0.1239

Optimal   pore size (ÎĽm)

0.2278

0.0386

Minimum   pore size (ÎĽm)

0.1592

0.0318

Average   pore size (ÎĽm)

0.2235

0.0377

In this study, the flux (J) of the membrane was obtained by permeating the solution through the membrane under a pressure of 0.1 MPa (1bar). We have corrected all the content and pictures about the flux in the text. And, the equation is corrected as follows:

                      (1)

To verify the flux of M2 membrane, we cautiously measure the flux change of the membrane within 120 minutes. The flux of M2 membrane is shown in Fig. 1(e-f). It can be seen that, at the beginning, the flux of membrane decrease slightly and then gradually stabilized. Moreover, the molecular configuration also affects the characterization of the molecular weight cut-off, and the rejection of the same molecular may be completely different. Our experiments could not measure molecular weight cut-off, but we think that molecular weight cut-off is closely related to the pore size. Therefore, we measure the pore size distribution of the membrane according to the bubble point and mean flow pore test. Fig. 1(e-f) illustrates the pore size distribution of LDH-M and M2 membranes. It can be seen that, the LDH-M membrane has a bigger average pore size than M2 membrane. The obtained results showed that the average pore size of M2 membrane was 0.0377 ÎĽm according to Table 1. Accumulation of the composite on the surface of membrane by layer-by-layer method could be main reason for the reduction of the average pore size. The reduction in the average pore size could increase the rejection rate of membrane. Moreover, due to the pore size decreases, the flux of the membrane also declines slightly. Nevertheless, the membrane flux could be related to the hydrophilic performance, construction method, pore size and other factors. The rejection experiments in the paper and the measure of membrane pore size have well reflected the filtration performance of the membranes. If you have better suggestions, please do not hesitate to contact us.

Point 2:The Figure 2(b1) shows that the LDH particles are deposited over the rod-like Sep layer. However, it was mentioned in section 2.2 that the Sep was deposited over the LDH layer. Explanation is needed here.

Response 2: Thanks for your suggestions. Our explanation is as follows:

In this study, we designed and manufactured a robust three-dimensional (3D) hierarchically rough sheetlike membrane, namely MgAl-LDH and Sep were evenly covered over CA membranes through layers and layers. The 3D structure uses the CA film as a substrate, LDH is selected as a 2D layer, and then Sep was introduced as a 3D layer. However, because of the strong electrostatic interaction and hydrogen bond between LDH and Sep, and the loose LDH, in some areas, LDH will break away from the original layer and insert into the layer of Sep. This phenomenon is also verified by AFM. Moreover, the smaller amount of Sep added results in a thinner Sep layer, so many LDH particles can be seen on the surface of the membrane in Figure 2(b1). With the increase of Sep, the Sep uniformly covered on the surface of membrane which displayed in Fig. S2 (b). Figure S2 (b) was shown in Supporting information. To avoid unnecessary misunderstanding, I have made a correction in the article.

Point 3:Figure 6: The variation of contact angle data can be due to variation of surface roughness and surface porosity of different samples. How did the authors consider these influential parameters?

Response 3: Thank you for your reminder. At first, the hydrophilicity of the material increase will reduce its contact angle. In this paper, the material has excellent hydrophilicity which is attributed to its abundant hydrophilic groups. Secondly, Quéré David et al. discuss the relationship between wettability and roughness [1]. For surfaces with a contact angle of less than 90 degrees (non-hydrophobic surfaces), the literature indicated that the surface roughness increase will reduce its contact angle. The AFM data herein demonstrates that the composite film roughness increase compared to the pure LDH-M film. Thirdly, the contact angle is related to the porosity and permeability of the membrane. The greater the porosity and permeability of the membrane, the measured contact angle will be smaller.

The literature is as follows:

[1] Quéré, David. Wetting and Roughness[J]. Annual Review of Materials Research, 2008, 38(1):71-99.

Point 4:It was claimed that the prepared composite membranes have high thermal stability, however, no experimental characterization is provided. How did the authors evaluate the thermal stability of the prepared membranes?

Response 4: Thank you for your professional comments. We are very sorry for our mistake. The thermally stability at the beginning of the article is a writing error. Our technology did not reach such experimental standard. In this study, we only studied the durability and stability of the membrane. The durability and stability of the filtration membrane was evaluated by cyclic experiments, contact angle measure in acid-base salt environments and immersion experiments. We have carefully revised the full manuscript and will no longer have similar writing errors. Thank you for your comments. If there is a chance in the future, I will study more about this.

Non-scientific comments:

Point 1:English (grammar, type, and punctuation) needs significant improvement. Few examples are:

-Page 1, Abstract, line 15: “is designing a highly efficient and flux material”

-Page 1, Abstract, line 18: “composite membranes hierarchical architecture via in 3D hierarchical architectures construction methods.”

-Page 10, line 252: “membranes also were low oil adhesiveness to underwater oil”

-Page 15, line 358: “membranes with the multifunction of anionic and cationic dyes”

Response 1: Thank you for your correction. We are sorry for our errors. We have carefully revised the full manuscript, and tried to improve the language level and avoid grammatical mistakes. All modifications will not change the original meaning.

Point 2:Page3, “scheme 1”should be named as “Figure 1”.The number of other figures need to be changed accordingly.

Response 2: Thank you, this section has been changed in the text. The order of the diagrams has also been changed accordingly.

Point 3:Page 5, line 175: The stability of the composite film cannot be evaluated based on SEM photograph. Revision is needed.

Response 3: Thank you for your professional advice. The SEM image herein is not used to illustrate the stability of the film. The photograph of the film simply indicates that the film is not easily fall off. In this study, we studied the durability and stability of the membrane. The durability and stability of the filtration membrane was evaluated by cyclic experiments, contact angle measure in acid-base salt environments and immersion experiments.

Point 4:Page 8, Fgure4, the range of the vertical aixs of the AFM images needs to be equal to allow easier comparison of the surface topography.

Response 4: Thank you for your comment. The range of the vertical aixs of the AFM images is equal. It's just that the starting point value of z is different.

Point 5:Figure 11, the Y-axis for rejection should be changed to 93 to 100 instead of 0.93 to 1.0. Same comment for Figure 13.

Response 5: Thank you for your correction. We have corrected the values on the Y-axis of the rejection rate in the figure.

Point 6:Figure 12 & Figure S4: Numbers should be presented with similar significant digits.

Response 6: Thank you for your reminder. This part has been corrected on the figure.

Point 7:Table S2: The surface roughness data is suggested to be presented in unit of "nm". Also, standard deviation needs to be reported for the data.

Response 7: Thank you for your thoughtful suggestions. The unit of the surface roughness data has been changed to "nm". The standard deviation has been added in Table S2 of the supporting information.

Round  2

Reviewer 1 Report

Authors revised the article significantly. It can be accepted for publication. 

Reviewer 2 Report

The authors have addressed all the comments appropriately. The manuscript can be published in the present form.